# Anatomical and Functional Changes of the Retina and the Choroid after Resolved Chronic CSCR

**DOI:** 10.3390/jcm8040474

**Published:** 2019-04-07

**Authors:** Lisa Toto, Rossella D’Aloisio, Rodolfo Mastropasqua, Luca Di Antonio, Marta Di Nicola, Giuseppe Di Martino, Federica Evangelista, Emanuele Erroi, Emanuele Doronzo, Cesare Mariotti

**Affiliations:** 1Ophthalmology Clinic, Department of Medicine and Science of Ageing, University G. d’Annunzio Chieti-Pescara, 66100 Chieti, Italy; l.toto@unich.it (L.T.); monsieurluca@yahoo.com (L.D.A.); federica.evan@hotmail.com (F.E.); erroi.emanuele@gmail.com (E.E.); doronzo.oculista@libero.it (E.D.); 2Vitreoretinal Unit, Bristol Eye Hospital, University of Bristol, Bristol, BS1 4TB, UK; rodolfo.mastropasqua@gmail.com; 3Eye Clinic, Polytechnic University of Marche, 60126 Ancona, Italy; mariottic@libero.it; 4Department of Medical, Oral and Biotechnological Sciences, Laboratory of Biostatistics, University “G. d’Annunzio” Chieti-Pescara, 66100 Chieti, Italy; marta.dinicola@unich.it; 5Department of Medicine and Science of Ageing, School of Hygiene and Preventive Medicine, University G. d’Annunzio Chieti-Pescara, 66100 Chieti, Italy; peppinodimartino@hotmail.com

**Keywords:** eplerenone therapy, chronic central serous chorioretinopathy, subfoveal choroidal thickness, optical coherence tomography angiography, superficial capillary plexus density, superficial capillary plexus density, choriocapillaris density

## Abstract

Background: To investigate anatomical/functional changes after oral eplerenone therapy for chronic central serous chorioretinopathy (CCSC) in successfully treated eyes and fellow eyes and assess timing of foveal subretinal fluid (SRF) resolution. Methods: Twenty-one eyes of 21 patients suffering from CCSC with monolateral foveal SRF successfully treated with oral eplerenone were enrolled in this retrospective study (group 1). The fellow eyes (21 eyes; group 2), healthy or affected by CCSC, without foveal SRF were considered in the analysis. A control healthy group was enrolled as well (healthy controls; *n* = 21). Main outcome measures during follow-up included changes of best corrected visual acuity (BCVA, logMAR), central macular thickness (CMT; µm), SRF (µm), subfoveal choroidal thickness (SFCT; µm), superficial capillary plexus density (SCPD, %), deep capillary plexus density (DCPD, %), and choriocapillaris density (CCD, %) and percentage of eyes showing foveal SRF resolution at different time points. Results: Functional and anatomical parameters significantly improved during the study in group 1. BCVA increased significantly (*p* < 0.001), while CMT, SFCT, and SRF decreased significantly (*p* < 0.001; *p* < 0.001, and *p* = 0.037, respectively). SCPD, DCPD, and CCD did not show any statistically significant difference during follow-up. In 71.4% of eyes, resolution of SRF was observed within 60 days and in the remaining 28.6%, at 120 days. In fellow eyes, SFCT decreased significantly (*p* < 0.001), whilst all other parameters did not modify. Conclusions: Eplerenone treatment in chronic CSCR potentially improves recovery of retinal and choroidal morphology as well as visual acuity gain. A complete resolution of foveal SRF was observed in all eyes during a 4-month follow-up, with most eyes healing at 2 months.

## 1. Introduction

Serous neurosensory retinal detachment and pigment epithelial detachment (PED) are typical features of central serous chorioretinopathy (CSC) that is a disease affecting young people, idiopathic or secondary in some cases to corticosteroid consumption or high levels of endogenous corticosteroid [1]. CSC has an estimated annual incidence of 1:10000, and its pathogenesis is still controversial [1].

The most common symptoms of CSC at the onset of the disease are blurred vision with the presence of a relative central scotoma and metamorphopsia, dyschromatopsia, micropsy, and reduction of contrast sensitivity [2].

Central serous chorioretinopathy can resolve spontaneously generally before 3 months from the onset, defining the acute form of the disease. A persistence of the neurosensory retinal detachment from 3 to 6 months characterizes the chronic form of CSC (CCSC), otherwise defined as diffuse retinal pigment epitheliopathy, that is usually associated with progressive alteration of the outer retinal layers and retinal pigment epithelium (RPE) with impairment of central visual function [2,3]. The choroidal circulation is involved in the pathogenetic mechanisms of CSC [4,5]. 

Increased hydrostatic pressure in the choroidal circulation may exceed the RPE pump and lead to a progressive accumulation of fluid in the subretinal space. The origin of increased vascular permeability of the choroid is still not clear. In CSC regulation of choroidal circulation seems to be influenced by some biochemical processes involving steroids, catecholamines, and sympathomimetic agents [4].

Treatment of the CSC condition includes different therapeutic possibilities, such as simple observation, laser treatment, photodynamic therapy (PDT) with verteporfin, intravitreal injection of anti-VEGF (Vascular Endothelial Growth Factor) drugs, subthreshold micropulse laser treatment, and treatment with antagonists of mineralocorticoid receptors (MR) [1].

Eplerenone is a commercially available mineralcorticoid specific receptor antagonist and is approved by US Food and Drug Administration for systemic conditions, such as heart failure and hypertension. In several studies, eplerenone has been used for CSC treatment both in the acute and chronic form of the disease showing high efficacy, particularly in CCSC [6,7,8,9]. 

We included retrospectively eyes with CCSC and monolateral foveal subretinal fluid (SRF) successfully treated with oral eplerenone evaluating modifications of functional and anatomical parameters in the diseased eyes and fellow eyes and assessing timing of foveal SRF resolution.

## 2. Experimental Section

### 2.1. Materials Study Population

A total of 42 eyes of 21 patients were analyzed in this retrospective observational study. Patients suffering from CCSC (nonresolving SRF for more than 4 months) with unilateral foveal involvement treated with oral eplerenone at a dose of 25 mg for the first week and then 50 mg for the following weeks between February 2017 and May 2018 and showing complete resolution of SRF were included (group 1; *n* = 21). The fellow eyes, healthy or affected by CCSC, without foveal SRF were considered in the analysis (group 2; *n* = 21). A control healthy group was enrolled as well (healthy controls; *n* = 21).

All patients had a diagnosis of CCSC with a persistence of foveal SRF for at least 4 months, documented by a clinical (indirect fundoscopy) and instrumental evaluation, including multicolor imaging, fundus autofluorescence (FAF), fluorescein angiography (FA), indocyanine green angiography, and spectral domain optical coherence tomography (SD OCT). In particular, the duration of CSCR was 6 months in six patients and 9–12 months in 15 patients.

Inclusion criteria were: Diagnosis of CCSC with unilateral persistent SRF involving the fovea for more than 4 months, which had been resistant to therapy with oral administration of acetazolamide and with no other previous treatment for CSC including focal laser therapy, PDT or anti-VEGF intravitreal treatment.

Exclusion criteria were: Other chorioretinal diseases, systemic contraindications for a mineralcorticoid specific receptor antagonist, surgical treatments within 6 months, including intravitreal injections. 

Best corrected visual acuity measured in LogMAR (BCVA), anterior segment biomicroscopy, intraocular pressure, indirect fundus exam, multicolor imaging, FAF, SD OCT, and OCT angiography (OCTA) were performed at baseline, at 7 days, and then monthly until SRF resolution after oral therapy. 

The study was conducted in accordance with Declaration of Helsinki, and our Institutional Review Board approved the retrospective chart review.

Written consent was obtained from all the participants. 

### 2.2. SD OCT Analysis

The acquisition protocol for SD OCT (Spectralis, HRA Heidelberg, Heidelberg, Germany) included a 49 horizontal raster dense linear B-scans centered on the fovea. Horizontal and vertical B-scans centered on the fovea with enhanced depth imaging (EDI) mode were acquired in all patients. 

All acquisitions following the baseline visit were acquired using the follow-up function. 

Central macular thickness (CMT) was measured using the central 1-mm diameter circle of the ETDRS (Early Treatment for Diabetic Retinopathy Study) thickness map. Subretinal fluid defined as the vertical distance between the end of the outer segment and the RPE at the foveal center was measured using the inbuilt manual caliper. 

Subfoveal choroidal thickness (SFCT), measured vertically from the outer border of the RPE to the inner border of the sclera, was measured using the inbuilt manual caliper on EDI OCT scans.

### 2.3. OCTA Analysis

#### 2.3.1. SD-OCT Angiography with XR Avanti and Vascular Layer Segmentation

SD-OCT (XR Avanti^®^; Optovue, Inc., Fremont, CA, USA) and OCTA with SSADA software (XR Avanti^®^ AngioVue) were performed in all the participants at baseline, 7 days, and monthly after eplerenone treatment. OCTA scans were acquired following a standardized protocol based on the SSADA algorithm (version 2017.1.0.144) as previously described [10]. 

Vascular retinal layers were visualized and segmented as previously described in the superficial capillary plexus (SCP), deep capillary plexus (DCP) and the mean choriocapillaris (CC) [10]. 

The projection-resolved algorithm was used to remove projection artifacts from the inner vascular plexus in the deep vascular plexus. This algorithm retains flow signals from blood vessels while suppressing projected flow signals in deeper layers. Images were reviewed by two investigators (L.T. and R.DA.) for segmentation accuracy; if segmentation errors were observed, then they were corrected using the segmentation and propagation tool from AngioVue (Angiovue, Optovue, Freemont, CA, USA). Final images were reviewed again to confirm segmentation placement in all B-Scans.

#### 2.3.2. Quantitative Vessel Analysis

Objective quantification of vessel density was carried out for each eye using SSADA software. A quantitative analysis was performed on the OCTA en-face images for each eye using AngioVue software as previously described [10]. 

Vessel densities of the SCP, DCP, and CC were evaluated in the foveal and parafoveal areas. Vessel density was defined as the percentage of the area occupied by vessels in a circular region of interest (ROI) of 3 mm in diameter positioned on the center of the foveal avascular zone and including the foveal area (1 mm of diameter) and the parafoveal area, which constitutes the remaining part inside the ROI. 

### 2.4. Main Outcome Measures

Main outcome measures during follow-up included changes of BCVA, CMT, SRF, SFCT, SCP density (SCPD), DCP density (DCPD) and CC density (CCD) in the diseased eyes (group 1), fellow eyes (group 2), and in healthy controls; percentage of eyes showing SRF resolution at different time points. 

### 2.5. Statistical Analysis

Quantitative variables were summarized as mean and standard deviation (SD) or median and interquartile range (IQR), according with their distribution. Categorical variables were summarized as frequency and percentage. Shapiro–Wilk’s test was performed to evaluate the departures from normality distribution for each variable.

The Kruskall–Wallis *H* test and chi-squared test were applied to evaluate statistically significant differences between groups (group 1, group 2, and healthy controls) for quantitative and qualitative parameters, respectively. Post-hoc a priori planned analysis was conducted to evaluate pairwise difference between groups.

A repeated-measures mixed model with linear trend analysis was performed to evaluate the effect of time (within factor), the effect of group (between factor), and their interaction (time × group), separately for each quantitative parameter. Contrast analysis was performed to evaluate differences of each measurement from baseline values. 

The Mann–Whitney *U*-test was performed to evaluate differences in absolute variation of each parameter between patients healed after 60 days and patients healed after 120 days of treatment. The Kaplan–Meier method was applied to estimate the resolution rate during follow-up. 

Statistical analysis was performed using IBM^®^ SPSS Statistics version 20.0 software (SPSS Inc., Chicago, IL, USA).

## 3. Results

The mean age was 48.7 ± 9.1 (range, 37–64) years, and they were 18 males (85.7%) and 3 females (14.3%). The clinical characteristics of the patients are summarized in Table 1.

At baseline, significant differences were found in terms of BCVA, CMT, and foveal SRF presence between group 1 and the two other groups of fellow eyes (group 2) and healthy controls (Table 1). 

SFCT and vessel density of all layers showed a significant difference between diseased eyes (group 1) and healthy controls (Table 1).

All diseased eyes (21) showed foveal SRF and subtle RPE alterations or widespread RPE atrophy. In particular, 15 eyes (71.4%) showed widespread RPE atrophy, and the remaining six eyes (28.6%) subtle RPE changes. In three eyes, there was a PED and in one eye, presence of intraretinal fluid. 

All fellow eyes except for three (18/21) presented RPE alterations with or without presence of extrafoveal fluid or PED. In detail, 12 eyes showed subtle RPE changes (57.1%), six eyes (28.6%) widespread RPE atrophy, and three eyes (14.3%) no RPE alterations. In nine eyes (42.9%), there was presence of SRF not involving the fovea. In six eyes (28.6%), there was a PED.

No active CNV was detected in any of the diseased and fellow eye. 

Overall, BCVA significantly increased during the treatment period in group 1 (*p* < 0.01; Table 2) and did not change in group 2.

In group 1, mean CMT changed significantly (*p* < 0.001; Table 2); by contrast, it did not change significantly in group 2, as shown in Table 2. 

Subretinal fluid was 217.14 ± 86.97 µm in group 1 at baseline (Figure 1A–C) and changed to 153.57 ± 120.49 at 30 days, 158.43 ± 119.53 at 60 days, and 134.71 ± 119.24 at 90 days (Figure 1B,C). Foveal subretinal fluid was completely absent at 120 days in all cases (Table 2, Figure 1B,C).

In detail, 3/21 eyes (14.3%) showed complete resolution of SRF at 1 month, 12/21 (57.1%) at 2 months, and 6/21 (28.6%) at 120 days. 

In the fellow eye, extrafoveal SRF completely disappeared in eight out of nine eyes.

Subfoveal choroidal thickness changed from 452.85 ± 132.74 µm at baseline to 386.85 ± 130.69 µm at 30 days, 354.85 ± 97.88 µm at 60 days, 358.57 ± 93.47 µm at 90 days, and 358.28 ± 93.33 µm at 120 days (*p* < 0.001) in group 1 and reduced significantly from 410.28 ± 124.19 µm at baseline to 394.42 ± 134.85 µm at 30 days, 374.14 ± 108.13 µm at 60 days, 374.28 ± 108.32 µm at 90 days, and 373.71 ± 107.62 µm at 120 days in group 2 (*p* < 0.001; Table 2).

The mean SCPD, the mean DCPD, and the mean CCD did not modify significantly during follow-up (Table 2; Figure 2).

In group 1 in the subanalysis, there were no significant differences in terms of anatomical and functional parameters between eyes with complete fluid resolution within 60 days and those within a 120-day resolution, as shown in Table 3. 

Complete fluid resolution rate at 60 and 120 days was, respectively, 72.4% (15 patients) and 100% (Figure 3). No treatment-related complications were observed in any patients. 

## 4. Discussion

In this study, we aimed at investigating eyes with CCSC and monolateral foveal SRF successfully treated with oral eplerenone, evaluating modifications of functional and anatomical parameters in the diseased eyes and fellow eyes and timing of foveal SRF resolution. We demonstrated a significant improvement of BCVA and a significant decrease of CMT, SRF, and SFCT during a 4-month follow-up in the diseased eyes. Retinal capillary density and choriocapillaris density did not show significant modifications during follow-up compared to baseline values. Complete resolution of SRF was observed in 14.3% of eyes at 1 month, in 57.1% at 2 months, and in the remaining 28.6% at 120 days. 

In the fellow eyes, SFCT significantly reduced, whilst all the other parameters did not modify significantly. 

It has been demonstrated in animal models that the neurosensorial retina, the RPE, and the choroid express MR, and it has been hypothesized that an overexpression/overactivation of MR is involved in the pathogenesis of CSC, thus suggesting a potential use of antagonists of MR, such as spironolactone and eplerenone, as a possible treatment in CSC patients [11,12,13].

In animal and clinical studies, oral MR-antagonists showed a pharmacological effect on CSC eyes [11,14,15,16,17].

Schwartz et al. found an excellent safety profile of eplerenone for treating CCSC, showing no antiandrogenic side effects and being more tolerable than spironolactone [6]. The latter has shown lots of drug-related complications due to its strong affinity to other steroid receptors [6].

Indeed, eplerenone was first used for CCSC in a nonrandomized pilot study showing total reabsorption of SRF in 9/11 eyes during a 3-month follow-up (three eyes with SRF reabsorption at 1 month and six eyes at 3 months) [16]. 

Subsequent studies demonstrated the efficacy of MR antagonists in the treatment of chronic and recurrent forms of CSC with functional (BCVA increase) and anatomical (SRF reabsorption and subfoveal choroidal thickness reduction) improvement [17,18,19,20].

In different studies, percentages of resolution of CCSC after eplerenone use have been reported, ranging from 13% to 31% at about two months and from 29% to 61% at about 4 months after treatment, respectively [7,8,9]. Complete reabsorption of foveal SRF has been reported, starting from the first month after treatment up to twelve months [9,16,21]. 

In our study, SRF decreased significantly, starting from the first month, and was completely absent at 120 days, and no signs of active CNV were found in any sample. 

In detail, 3/21 eyes (14.3%) showed complete resolution of SRF at 1 month, 12/21 (57.1%) at 2 months, and 6/21 (28.6%) at 120 days. We can speculate that these successful results can be related to the retrospective nature of this work. Indeed, we considered retrospectively only cases that showed complete resolution of SRF and evaluated time of resolution and changes during treatment. Moreover, the relatively short duration of the disease of some patients in our sample (6 months in six patients) could be another possible reason for treatment success.

Likewise, Sacconi et al. observed that a complete response to the oral therapy was associated with absence of CNV [7]. The latter has been considered a possible predictive biomarker of treatment response [7].

Among patients with SRF resolution at 4 months, there was one patient with intraretinal fluid at baseline and six patients with widespread retinal atrophy. As already observed in literature, RPE changes have been associated with a slower therapy response with eplerenone [6,7,8], confirming our findings. 

Indeed, Cakir et al. identified the presence of widespread RPE and ellipsoid zone disruption as possible response predictors to eplerenone therapy in CCSC patients resistant to other treatments, with a tendency towards a lower gain in visual acuity [8]. 

Similarly to other studies, we reported SFCT reduction after treatment in diseased eyes, confirming a role of the choroid in the pathogenesis of the disease. 

Interestingly, in our series, there was no difference of SFCT at baseline between the diseased eyes and fellow eyes, and SFCT of the fellow eyes showed a significant reduction during treatment as well. It is known that patients with CSC are characterized by an increased choroidal thickness both in diseased eyes and fellow eyes if compared with age-matched normal eyes, likely due to choriocapillaris hyperpermeability [22,23,24], and it has been suggested that the occurrence of choroidal thickening in the unaffected contralateral eye could represent a precursor to retinal detachment [23]. Maruko et al. found no significant difference in terms of thickness increase of SFCT in the fellow eyes between acute and chronic forms of CSC [23].

In our work, some of the fellow eyes showed subtle or widespread RPE alterations associated or not with SRF not involving the fovea that could be the evidence of a bilateral disease that is more frequent in the chronic forms, suggesting a possible pre-existing CSC condition.

The reduction of SFCT also in the fellow eye is a benefit of the systemic treatment, thus probably lowering the risk of foveal retinal detachment or treating bilaterally the condition if SRF is present in the fellow eye.

As recently reported, oral eplerenone treatment does not affect choriocapillaris density, similarly to our findings [6,25]. 

Some studies have reported vessel density modifications mainly of DCPD, probably related to mechanical displacement of retinal vessels caused by SRF. In CCSC eyes, due to the pathogenesis of the disease typically involving choroid and outer retinal layers, DCP vessel density reduction after eplerenone was not statistically significant [25]. Similarly, OCTA analysis in our sample showed no chorioretinal vascular changes after the systemic therapy.

In summary, eplerenone treatment in chronic CSCR potentially improves recovery of retinal and choroidal morphology as well as visual acuity gain.

The main limitations of our study are its retrospective nature and the small sample. A larger sample is needed to provide more information about functional and anatomical changes and to evaluate accurately the possible systemic effect of the oral mineralocorticoid receptor antagonists.

## Figures and Tables

**Figure 1 jcm-08-00474-f001:**
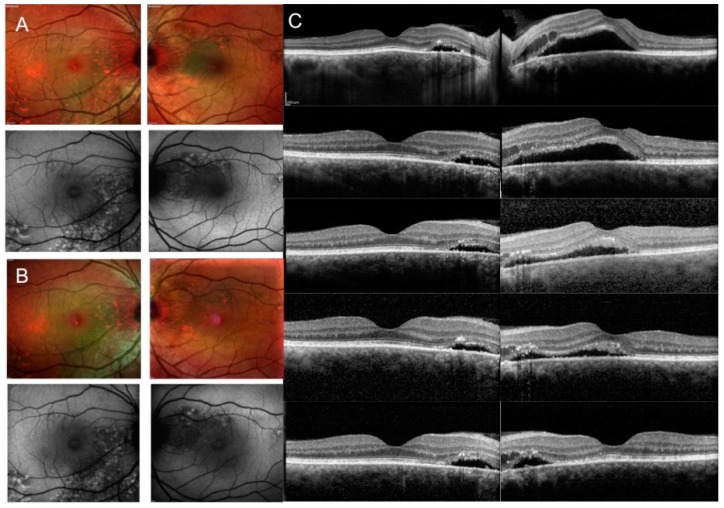
Multimodal imaging of patients with chronic central serous chorioretinopathy (CCSC) at baseline and during a 4-month follow-up during eplerenone treatment. Multicolor fundus image (MFI) (first row) and fundus autofluorescence (FAF) (second row) of right and left eye images showing widespread retinal pigment epithelium (RPE) atrophy in both eyes and foveal subretinal fluid (SRF) in left eye at baseline (**A**); MFI and FAF of both eyes after SRF resolution at 4 months (**B**); spectral domain optical coherence tomography (SD OCT) of right and left eye showing extrafoveal SRF in right eye and foveal SRF and extrafoveal intraretinal fluid in left eye at baseline (first row) and foveal SRF resolution during follow-up in left eye (30 days, second row; 60 days, third row; 90 and 120 days, fourth and fifth rows, respectively) (**C**).

**Figure 2 jcm-08-00474-f002:**
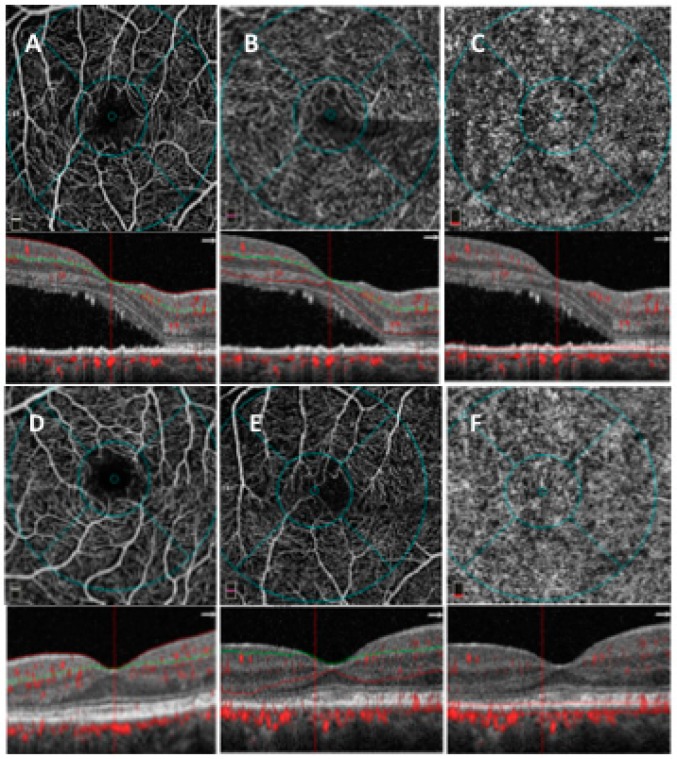
OCT angiography (OCTA) images of patients with CCSC at baseline and at a 4-month follow-up during eplerenone treatment. OCTA images at baseline of superficial (**A**), deep (**B**), and choriocapillaris (**C**) layers (first row). OCTA images after treatment at 4-month follow-up of superficial (**D**), deep (**E**), and choriocapillaris (**F**) layers (second row).

**Figure 3 jcm-08-00474-f003:**
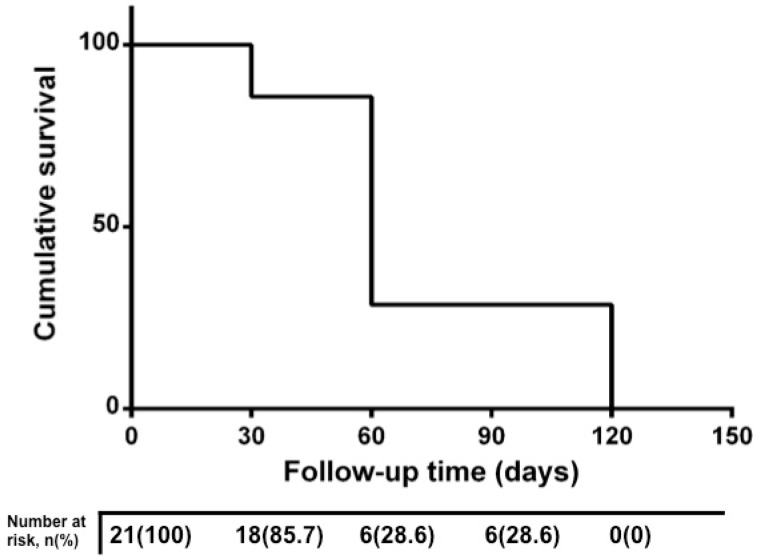
Kaplan–Meier curve of resolution rate during follow-up. Kaplan–Meier curve of resolution rate showing the timing of subretinal fluid reabsorption during 4-month follow-up.

**Table 1 jcm-08-00474-t001:** Ocular characteristics of diseased eyes (group 1), fellow eyes (group 2) and healthy controls.

Ocular Characteristics	Group 1 (*n* = 21)	Group 2 (*n* = 21)	Healthy Controls (*n* = 21)	Kruskall–Wallis *p*-Value	Post-hoc *p*-Value Group 1 vs. Group 2
Baseline visual acuity (logMAR), mean ± SD	0.30 ± 0.11 **	−0.01 ± 0.04	0.04 ± 0.08	<0.001	<0.001
Foveal SRF (µm), mean ± SD	217.14 ± 86.97	-	-		
CMT (µm), mean ± SD	428.71 ± 80.05 **	237.71 ± 37.56	248.86 ± 17.25	<0.001	<0.001
SFCT (µm), mean ± SD	452.85 ± 132.74 **	410.28 ± 124.19 **	252.00 ± 53.39	<0.001	0.387
Foveal SRF, *n* (%)	21 (100.0)	-	-		
Only extrafoveal SRF, *n* (%)	-	9 (42.8)	-		
Subtle RPE changes/widespread RPE atrophy, *n*	6/15	12/6	-	0.017 ^b^	
PED, *n* (%)	3 (14.3)	6 (28.6)	-	0.259 ^a^	
Intraretinal fluid, *n* (%)	1 (4.8)	-	-		
SCPD (%), mean ± SD	46.78 ± 4.29 **	45.04 ± 4.52 *	51.08 ± 3.00	<0.001	0.869
DCPD (%), mean ± SD	46.34 ± 5.36 **	46.81 ± 4.66 **	58.68 ± 3.03	<0.001	0.989
CCD (%), mean ± SD	54.72 ± 4.93 **	61.50 ± 5.58 **	66.70 ± 1.19	<0.001	<0.001

^a^ Fisher exact test group 1 vs. group 2; ^b^ chi-square test group 1 vs. group 2; * *p* < 0.05 ** *p* < 0.01 post-hoc analysis vs. healthy controls group. SRF, subretinal fluid; CMT, central macular thickness; SFCT, subfoveal choroidal thickness; RPE, retinal pigment epithelium; PED, pigment epithelial detachment.

**Table 2 jcm-08-00474-t002:** Functional and morphological parameters at baseline and after eplerenone therapy in two groups.

	Baseline	7 days	30 days	60 days	90 days	120 days	*p*-Value ^a^	*p*-Value ^b^	*p*-Value ^c^
BCVA, logMAR							<0.001	<0.001	<0.001
Group 2	−0.01 ± 0.04	−0.01 ± 0.03	−0.01 ± 0.03	0.02 ± 0.07	0.02 ± 0.07	0.02 ± 0.07			
Group 1	0.30 ± 0.11	0.30 ± 0.11	0.13 ± 0.11 **	0.11 ± 0.10 **	0.10 ± 0.08 **	0.09 ± 0.09 **			
CMT, µm							<0.001	<0.001	<0.001
Group 2	237.71 ± 37.56	223.28 ± 16.71	231.57 ± 27.48	223.85 ± 19.99	224.14 ± 19.60	224.14 ± 19.60			
Group 1	428.71 ± 80.05	427.86 ± 110.53	289.57 ± 80.26 **	267.71 ± 89.07 **	251.71 ± 63.46 **	213.57 ± 32.80 **			
SRF, µm							0.037 ^d^	-	
Group 2									
Group 1	217.14 ± 86.97	219.42 ± 112.98	153.57 ± 120.49 **	158.43 ± 119.53 **	134.71 ± 119.24 **	0 **			
SFCT, µm							<0.001	0.939	0.452
Group 2	410.28 ± 124.19	392.28 ± 135.22	394.42 ± 134.85	374.14 ± 108.13	374.28 ± 108.32	373.71 ± 107.62			
Group 1	452.85 ± 132.74	427.71 ± 125.85	386.85 ± 130.69 **	354.85 ± 97.88 **	358.57 ± 93.47 **	358.28 ± 93.33 **			
SCPD, µm							0.354	0.853	0.368
Group 2	45.04 ± 4.52	45.90 ± 5.75	48.06 ± 2.63	48.02 ± 4.06	47.48 ± 3.24	47.94 ± 3.58			
Group 1	46.78 ± 4.29	46.76 ± 5.91	47.05 ± 4.55	47.05 ± 4.55	47.64 ± 4.48	47.68 ± 4.52			
DCPD, µm							0.212	0.832	0.565
Group 2	46.81 ± 4.66	47.59 ± 5.66	47.40 ± 3.58	48.07 ± 3.16	48.41 ± 3.39	48.58 ± 2.85			
Group 1	46.34 ± 5.36	48.86 ± 5.51	45.37 ± 4.31	46.57 ± 4.46	46.60 ± 4.61	47.44 ± 3.53			
CCD, µm							0.452	0.417	0.358
Group 2	61.50 ± 5.58	60.92 ± 6.18	61.05 ± 6.31	61.35 ± 5.81	61.88 ± 5.21	62.84 ± 5.13			
Group 1	54.72 ± 4.93	55.87 ± 4.96	59.57 ± 5.91	60.84 ± 6.08	61.02 ± 4.54	61.28 ± 4.81			

Data are expressed as mean ± standard deviation. Probability that effect of treatment on the addressed variable is influenced by: ^a^ time: For each variable, the differences have been tested between the means at each time point of the two groups. ^b^ groups: For each variable, the differences have been tested between the means of group 1 over the time and the means of the group 2 over time. ^c^ interaction time × group: Probability that the effects of treatment are greater in one distinct group. ^d^ Analyses performed only among cases; * *p* < 0.05. ** *p* < 0.01 pairwise post-hoc analysis vs. baseline measurement. BCVA, best corrected visual acuity; CMT, central macular thickness; SRF, subretinal fluid; SCPD, superior capillary plexus density; DCPD, deep capillary plexus density; CCD, choriocapillaris density.

**Table 3 jcm-08-00474-t003:** Differences in absolute variation between patients with SRF reabsorption after treatment within 60 days and within 120 days expressed as median and interquartile range (IQR).

Variable	60 days *n* = 15	120 days *n* = 6	Mann–Whitney *U* test *p*-Value
BCVA, logMAR	−0.05 (−0.20–0)	−0.10 (−0.28–0)	0.566
CMT, µm	−81.00 (−184.00–1.75)	−125.00 (−252.00–4.50)	0.901
SFCT, µm	−46.00 (−74.7–3.25)	−102.00 (−212.00–56.00)	0.055
SCPD, µm	0.40 (−0.50–3.20)	2.00 (−0.18–3.80)	0.521
DCPD, µm	1.60 (0.60–2.90)	1.40 (0.88–3.80)	0.765
CCD, µm	4.00 (0.35–10.00)	1.90 (0.80–4.70)	0.547

BCVA, best corrected visual acuity; CMT, central macular thickness; SFCT, subfoveal choroidal thickness; SCPD, superior capillary plexus density; DCPD, deep capillary plexus density; CCD, choriocapillaris density.

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
