# Peer review of "Anatomical and Functional Changes of the Retina and the Choroid after Resolved Chronic CSCR"

_jcm, 2019, doi:10.3390/jcm8040474_

Round 1
Reviewer 1 Report
The authors investigated retrospectively the anatomical/functional changes after oral eplerenone therapy for 21 eyes with chronic central serous chorioretinopathy (CCSC). They assessed timing of foveal subretinal fluid (SRF) resolution in the affected eye (group 1). The 21 fellow eyes without foveal SRF were served as controls.
They present the following observations;
1) Functional and anatomical parameters significantly improved during the study in group 1.
2) BCVA increased significantly (p<0.001), while CMT, SFCT and SRF decreased significantly (p<0.001; p<0.001 and p=0.037, respectively).
3) SCPD, DCPD and CCD did not show any statistically significant difference during follow-up.
4) In 71.4% of eyes resolution of SRF was observed within 60 days and in the remaining 28.6% at 120 days.
5) In fellow eyes SFCT decreased significantly (p<0.001) whilst all other parameters did not modify.
They conclude that
1) Eplerenone was an effective treatment for CCSC showing recovery of retinal and choroidal morphology related to visual acuity improvement.
2) A complete resolution of foveal SRF was observed in all eyes during a 4-month follow-up with most eyes healing at 2 months.
The design is simple, the results are interesting, the message is straightforward. The study has important points and clinically relevant, but few points deserve additional consideration.
Major point
Authors say that this is retrospective study. However, ‘consecutive’ and ‘Experimental Section’ sound that this is more likely prospective study.
Minor points
1. Line 26, ‘subretinal fluid (SRF)’ may be ‘SRF’
2. Line 53, Authors mention that ‘REP’ shows ‘retinal pigment epitheliopathy’. But afterwards, ‘RPE’ seems to show ‘retinal pigment epithelium’.
3. Line 77, ‘subretinal fluid’ may be ‘SRF’
4. Line 82, ‘ICGA’ never appear afterwards. The definition for the abbreviation is unnecessary.
5. Line 91, ‘IOP’ never appear afterwards. The definition for the abbreviation is unnecessary.
6. Line 200, “In group 1 in the sub-analysis there were no significant differences in terms of anatomical and functional parameters between eyes with complete fluid resolution within 60 days and those within 120-day resolution, as showed in Table 3.” It seems that the authors compared 15 eyes where SRF resolved within 60 days and 21 eyes where SRF resolved within 120 days. Did the latter 21 eyes include the former 15 eyes? I am not sure it makes sense. Anyway, authors may show the sample number in the table 3 as, for example, ‘60 days (n=15)’ and ‘120 days (n=21)’ in the variable.
7. Although negative results, representative OCTA images with vessel density in each layer can be shown.
Author Response
The design is simple, the results are interesting, the message is straightforward. The study has important points and clinically relevant, but few points deserve additional consideration.
Major point
Authors say that this is retrospective study. However, ‘consecutive’ and ‘Experimental Section’ sound that this is more likely prospective study.
Thank you for your valuable comment. Our work was a retrospective chart review including patients with a diagnosis of CCSC with monolateral persistence of foveal SRF that were successfully treated with eplerenone. Thus we considered retrospectively only cases that showed complete resolution of SRF and evaluated time of resolution and changes during treatment.
All clinical (indirect fundoscopy) and instrumental examinations, including multicolor imaging, FAF, FA, ICGA, SD OCT and all follow-up controls were the routinely management of patients with this type of disease referring to our medical retina centre. The follow up control at 7 days was set to check treatment tolerance and to increase the dosage and each month to follow the course of the disease.
We have delated the word ‘consecutive’.
Minor points
1. Line 26, ‘subretinal fluid (SRF)’ may be ‘SRF’
We have checked and modified it.
2. Line 53, Authors mention that ‘REP’ shows ‘retinal pigment epitheliopathy’. But afterwards, ‘RPE’ seems to show ‘retinal pigment epithelium’.
We have checked and delated it.
3. Line 77, ‘subretinal fluid’ may be ‘SRF’
We have checked and modified it.
4. Line 82, ‘ICGA’ never appear afterwards. The definition for the abbreviation is unnecessary.
We have checked and delated ICGA abbreviation as suggested.
5. Line 91, ‘IOP’ never appear afterwards. The definition for the abbreviation is unnecessary.
We have checked and delated IOP abbreviation as suggested.
6. Line 200, “In group 1 in the sub-analysis there were no significant differences in terms of anatomical and functional parameters between eyes with complete fluid resolution within 60 days and those within 120-day resolution, as showed in Table 3.” It seems that the authors compared 15 eyes where SRF resolved within 60 days and 21 eyes where SRF resolved within 120 days. Did the latter 21 eyes include the former 15 eyes? I am not sure it makes sense. Anyway, authors may show the sample number in the table 3 as, for example, ‘60 days (n=15)’ and ‘120 days (n=21)’ in the variable.
Table 3 has been updated with the number of patients.
7. Although negative results, representative OCTA images with vessel density in each layer can be shown.
As suggested, representative OCTA images with vessel density in each layer have been added (Figure 2).
Reviewer 2 Report
General remarks:
Interesting study, however not novel according to available medical literature.
Authors have to decide whether they want to show functional and morphological impairment after chronic CSCR or simple present results of eplerenone treatment of chronic CSCR. Results are very optimistic, which is not in consent with other studies.
The material is valuable but should be presented in a different way. The easiest way to use the same material is to build a control group based on healthy individuals and make a comparison (both groups from the study could be used – one treated as active chronic CSCR, the other as partially resolved CSCR group or simply fellow eye group).
Specific remarks for the authors.
1. Abstract :
a. it has to be clarified what specific parameters were measured in CMT, SRF, SFCT, SCPD DCPD and CCD. Also it has to be pointed out how was BCVA measured (Snellen or LogMAR or ETDRS letters). The same applies to results – it has to be clarified which of the parameters have changed.
b. Criterium of chronicity of CSCR has to be depicted.
2. The title should be rephrased: the study shows functional and morphological changes of the RPE and choroid after resolved chronic CSCR. Eplerenone therapy has nothing to do with that change. Example: Anatomical and functional changes of the retina and the choroid after resolved chronic CSCR.
3. Introduction: subthreshold micropulse laser treatment has to be mentioned as a treatment method of CSCR.
4. Methods:
a. Control group should consist of healthy individuals with age distribution similar to the study group. Fellow eyes of CSCR patients are often eyes that had CSCR that resolved. ( Authors themselves reveal the presence of foveal RPE alterations, SRF or PED in control group ) Besides eyes with non-foveal location of SRF may have some secondary foveal changes. Generally speaking eyes of patients with CSCR might have different choroidal and retinal morphology if compared to healthy individuals (ex pachychoroid). That has been proved in other studies and has to be considered when constructing a control group. Thus , comparing the study group with the group consisting of the fellow eyes does not show a real impairment of the retinal function and morphology, as the fellow eyes had already been affected. For example in long standing but resolved CSCR significant retinal thinning is observed.
b. Oral therapy with acetazolamide is not a standard therapy of CSCR and its efficacy is not confirmed in available studies – the phrase has to be corrected
c. Main outcome measures : as in the abstract, authors should describe more precisely what parameters were measured.
d. OCTA measurements have to be described in more detail (Optovue software)
5. Results
a. Table 1 : I understand that this table analyzes the difference between the study group and the control group – this information has to be placed in the table title /subtitle. Does p value show statistical significance of that difference – it has to be clarified.
b. The study group has to be characterized and analyzed according to duration of CSCR. It is not enough to state that CSCR lasted longer than 4 months. Patient with duration of CSCR of 3-4 months is more likely to have spontaneous resolution of CSCR than the patient with longer duration of the disease.
c. It has to clarified that group 1 is the study group and group 2 is the control group (however I would strongly recommend not to use fellow eyes as the controls)
d. Description of results should not repeat data from the table, unless authors want to point out some specific facts.
e. Table 2 : OCTA measurements – what units were used to measure vessel density ? mm ? Please amend. Legend for table 2 has to be simplified. It has to be described what p – values mean.
f. It is hard to believe that 100% of SRF resolution was achieved after 120 days since the beginning of treatment. It is unusual for really chronic cases. This fact has to be elaborated more in discussion section. Again, it might be due to relatively short duration of CSCR ( a few months not years for example).
6. Discussion:
a. It is not certain whether the resolution of SRF is due to eplerenone therapy or spontaneous, especially as duration of CSCR is not taken into consideration. This fact has to be stressed.
b. Results of treatment have to be referred to other studies in detail (% of patients with complete SRF resolution).
7. The paper needs native speaker proofreading.
Author Response
General remarks:
Interesting study, however not novel according to available medical literature.
Authors have to decide whether they want to show functional and morphological impairment after chronic CSCR or simple present results of eplerenone treatment of chronic CSCR. Results are very optimistic, which is not in consent with other studies.
The material is valuable but should be presented in a different way. The easiest way to use the same material is to build a control group based on healthy individuals and make a comparison (both groups from the study could be used – one treated as active chronic CSCR, the other as partially resolved CSCR group or simply fellow eye group).
A control group has been provided and statistical differences compared to group 1 and group 2 at baseline have been reported in table 1.
Specific remarks for the authors.
1. Abstract:
a. it has to be clarified what specific parameters were measured in CMT, SRF, SFCT, SCPD DCPD and CCD. Also it has to be pointed out how was BCVA measured (Snellen or LogMAR or ETDRS letters). The same applies to results – it has to be clarified which of the parameters have changed.
Unit of measure of all parameters have been added in the abstract including BCVA that was measured in LogMAR and this detail has been added into the manuscript in Experimental section.
b. Criterium of chronicity of CSCR has to be depicted.
Criterium of chronicity of CSCR has already described in Experimental section: ‘Patients suffering from CCSC (non-resolving SRF for more than 4 months)’.
However, as suggested we added it in inclusion criteria as well.
2. The title should be rephrased: the study shows functional and morphological changes of the RPE and choroid after resolved chronic CSCR. Eplerenone therapy has nothing to do with that change. Example: Anatomical and functional changes of the retina and the choroid after resolved chronic CSCR.
We have changed the title as suggested.
3. Introduction: subthreshold micropulse laser treatment has to be mentioned as a treatment method of CSCR.
Subtreshold laser treatment has been mentioned as suggested.
4. Methods:
a. Control group should consist of healthy individuals with age distribution similar to the study group. Fellow eyes of CSCR patients are often eyes that had CSCR that resolved. (Authors themselves reveal the presence of foveal RPE alterations, SRF or PED in control group) Besides eyes with non-foveal location of SRF may have some secondary foveal changes. Generally speaking eyes of patients with CSCR might have different choroidal and retinal morphology if compared to healthy individuals (ex pachychoroid). That has been proved in other studies and has to be considered when constructing a control group. Thus, comparing the study group with the group consisting of the fellow eyes does not show a real impairment of the retinal function and morphology, as the fellow eyes had already been affected. For example in long standing but resolved CSCR significant retinal thinning is observed.
We added a control group of healthy individuals and statistical differences compared to group 1 and group 2 at baseline have been reported in table 1.
b. Oral therapy with acetazolamide is not a standard therapy of CSCR and its efficacy is not confirmed in available studies – the phrase has to be corrected
The word ‘conventional’ has been removed.
c. Main outcome measures: as in the abstract, authors should describe more precisely what parameters were measured.
Unit of measure of all parameters have been added in the abstract including BCVA that was measured in LogMAR and this detail has been added into the manuscript in Experimental section.
d. OCTA measurements have to be described in more detail (Optovue software).
OCTA scans were acquired following a standardized protocol based on the SSADA algorithm (version 2017.1.0.144) that is previously described and the reference has been provided.
5. Results
a. Table 1: I understand that this table analyzes the difference between the study group and the control group – this information has to be placed in the table title /subtitle. Does p value show statistical significance of that difference – it has to be clarified.
Table 1 shows differences between the diseased eyes (study group), the fellow eyes and the healthy control group that has been added in the work and in the analysis.
As already described on line 156-157, at baseline significant differences were found between the group 1 and the two other groups (group 2 and healthy controls) in terms of visual acuity that was worst in the study group as expected due to the presence of foveal subretinal fluid. In the study group CMT was statistically higher than both two as well. The p of significance is shown in the table.
b. The study group has to be characterized and analyzed according to duration of CSCR. It is not enough to state that CSCR lasted longer than 4 months. Patient with duration of CSCR of 3-4 months is more likely to have spontaneous resolution of CSCR than the patient with longer duration of the disease.
The duration of CSCR was 6 months in 6 patients and 9-12 months in 15 patients. This information has been added in the manuscript.
A subanalysis considering also the duration of CSCR was not possible for the small sample of patients.
c. It has to clarified that group 1 is the study group and group 2 is the control group (however I would strongly recommend not to use fellow eyes as the controls)
We have followed your recommendation and we have included a control group in the statistical analysis, in addition to the fellow eye.
d. Description of results should not repeat data from the table, unless authors want to point out some specific facts.
I have delated some sentences to not repeat data from the tables.
e. Table 2: OCTA measurements – what units were used to measure vessel density? mm ? Please amend. Legend for table 2 has to be simplified. It has to be described what p – values mean.
Vessel density was measured as percentages as shown in table 2.
f. It is hard to believe that 100% of SRF resolution was achieved after 120 days since the beginning of treatment. It is unusual for really chronic cases. This fact has to be elaborated more in discussion section. Again, it might be due to relatively short duration of CSCR (a few months not years for example).
Our work was a retrospective chart review including patients with a diagnosis of CCSC with monolateral persistence of foveal SRF that were successfully treated with eplerenone. Thus we considered retrospectively only cases that showed complete resolution of SRF and evaluated time of resolution and changes during treatment. Moreover as you correctly said, the relatively short duration of the disease of some patients (6 months in 6 patients) could be a possible reason for this fully treatment success taking into consideration the small sample of this work.
This interesting aspect has been added in discussion section.
6. Discussion:
a. It is not certain whether the resolution of SRF is due to eplerenone therapy or spontaneous, especially as duration of CSCR is not taken into consideration. This fact has to be stressed.
15 patients had 6-12 months of CSCR duration and this information has been provided in the manuscript.
b. Results of treatment have to be referred to other studies in detail (% of patients with complete SRF resolution).
This aspect has already been reported in discussion section:
‘In different studies percentages of resolution of CCSC after eplerenone use have been reported ranging from 13% to 31% at about two months and from 29% to 61% at about 4 months after treatment respectively [7-9]’.
7. The paper needs native speaker proofreading.
The paper has been accurately checked by an English native speaker.
Round 2
Reviewer 2 Report
The authors have to put the stress on the fact , that the study does not analyse effects of eplerenone therapy as only resolved cases are analysed. The study presents potential improvement in chronic CSCR treated by eplerenone with the assumption that the effect is not spontaneous but can be attributed to the drug effect.
I suggest change in the conclusion section of the abstract. Ex :
Eplerenone treatment in chronic CSCR potentially improves recovery of retinal and choroidal morphology as well as visual acuity gain.
I would also suggest a few sentences describing such approach in the end of discussion section of the manuscript.
Author Response
The authors have to put the stress on the fact , that the study does not analyse effects of eplerenone therapy as only resolved cases are analysed. The study presents potential improvement in chronic CSCR treated by eplerenone with the assumption that the effect is not spontaneous but can be attributed to the drug effect.
I suggest change in the conclusion section of the abstract. Ex :
Eplerenone treatment in chronic CSCR potentially improves recovery of retinal and choroidal morphology as well as visual acuity gain.
I would also suggest a few sentences describing such approach in the end of discussion section of the manuscript.
Thank you for your valuable comment. As suggested, we have changed the conclusion section of the abstract and we have stressed this aspect in the discussion section as well.